# Community health workers' efforts to build health system trust in marginalised communities: a qualitative study from South Africa

Jocelyn Anstey Watkins [iD],[1] Frances Griffiths [iD],[1,2] Jane Goudge [iD] [2]

[1]Division of Health Sciences, Warwick Medical School, University of Warwick, Coventry, UK
[2]Centre for Health Policy, School of Public Health, University of the Witwatersrand, Johannesburg, Gauteng, South Africa

**Correspondence to**
Prof Jane Goudge;
Jane.goudge@gmail.com

## ABSTRACT

**Introduction** Community health workers (CHWs) enable marginalised communities, often experiencing structural poverty, to access healthcare. Trust, important in all patient–provider relationships, is difficult to build in such communities, particularly when stigma associated with HIV/AIDS, tuberculosis and now COVID-19, is widespread. CHWs, responsible for bringing people back into care, must repair trust. In South Africa, where a national CHW programme is being rolled out, marginalised communities have high levels of unemployment, domestic violence and injury.

**Objectives** In this complex social environment, we explored CHW workplace trust, interpersonal trust between the patient and CHW, and the institutional trust patients place in the health system.

**Design, participants, setting** Within the observation phase of a 3-year intervention study, we conducted interviews, focus groups and observations with patients, CHWs, their supervisors and, facility managers in Sedibeng.

**Results** CHWs had low levels of workplace trust. They had recently been on strike demanding better pay, employment conditions and recognition of their work. They did not have the equipment to perform their work safely, and some colleagues did not trust, or value, their contribution. There was considerable interpersonal trust between CHWs and patients, however, CHWs' efforts were hampered by structural poverty, alcohol abuse and no identification documents among long-term migrants. Those supervisors who understood the extent of the poverty supported CHW efforts to help the community. When patients had withdrawn from care, often due to nurses' insensitive behaviour, the CHWs' attempts to repair patients' institutional trust often failed due to the vulnerabilities of the community, and lack of support from the health system.

**Conclusion** Strategies are needed to build workplace trust including supportive supervision for CHWs and better working conditions, and to build interpersonal and institutional trust by ensuring sensitivity to social inequalities and the effects of structural poverty among healthcare providers. Societies need to care for everyone.

## INTRODUCTION

Community health workers (CHWs) serve a critical function in low-income and middle-income countries, providing frontline services to marginalised groups who face significant barriers to care.[1–3] Effective deployment of CHWs is crucial to moving towards the Sustainable Development Goals (SDG).[4 5] In settings where HIV and tuberculosis (TB) are established epidemics, CHWs can assist people in adhering to treatment, important where drug resistance to antiretroviral treatment and TB are public health concerns.[6 7]

In vulnerable communities, where people experience structural poverty,[8] CHWs have to navigate complex health and social situations. Trust, important in all patient–provider–health system relationships,[9] is more fragile in such communities.[10] Patients who have unstable lives often receive poorer care;

in turn, poor quality care may cause patients to lose trust in their local facility, and become reluctant to seek care in the future.[11] CHWs, responsible for bringing people back into care, must repair that trust, a complex task when stigma associated with HIV/AIDS, TB and now COVID-19,[12 13] is widespread.

In South Africa, a national CHW programme is being rolled out, particularly in marginalised communities with high levels of unemployment,[14] domestic violence[15] and injury,[16] an epidemic fuelled by high rates of alcohol abuse.[17] There is also a high prevalence of communicable and non-communicable conditions.[18 19] Patients struggle to stay in care due to poverty[20] and stigma,[21 22] and those who are migrants face inequity in the health system.[23] CHWs themselves are fighting for employment rights and recognition of their contribution, and so are challenging their relationship with the health system.[24] In this complex social environment, we explored CHW workplace trust, interpersonal trust between the patient and CHW and the institutional trust patients place in the health system.

## BACKGROUND
### CHW programme in South Africa

In 2011, the South African Department of Health, initiated a national CHW programme (ward-based outreach teams; WBOTs) to improve access to services.[25] The intention is to provide health promotion, prevention, screening services and referral for a wide range of health and social needs.[26] The teams are composed of CHWs, supervised by one or two nurses, usually either a retired senior nurse (called a professional nurse) or junior nurse (called an enrolled nurse).[27] Professional nurses (PNs) in South Africa can diagnose patients, prescribe treatment and dispense medication. PN supervisors are trained in primary healthcare and community nursing. Enrolled nurses (ENs) complete a 2-year nursing course and are qualified to provide nursing care under supervision.

CHWs are lay people, members of the community. Prior to the national programme, there were a wide range of CHW programmes managed by a patchwork of non-governmental organisations, who in turn were often funded by government, that had emerged in the 1990s due to the HIV epidemic. These CHWs were transferred into the government programme as it started in 2011; there was no additional recruitment process. The CHWs underwent Phase 1 and Phase 2 standardised training to gain a nationally recognised certificate. This covers identification of the need for antenatal and postnatal care, monitoring immunisation and adherence to chronic medication, screening for malnutrition and TB, substance abuse and gender-based violence.[28 29] CHWs conduct household registrations to identify those in need, and trace patients who have withdrawn from care. During the COVID-19 pandemic, CHWs have been responsible for a mass community-based screening programme, asking people about symptoms and referring them to mobile testing units to quarantine suspected cases and provide appropriate care.[30]

### Trust

Trust, a complex, multifaceted notion, 'influences individuals' willingness to act on the basis of words, motives, intentions, actions and decisions of others under conditions of uncertainty, risk or vulnerability'.[31] The existing definitions and theoretical frameworks have done much to elucidate the ambiguity of trust.[32] The social relations of trust are accepted as a core contributor to health systems; a trust-based health system is grounded in cooperation, communication and empathy, enabling the successful functioning of the health service.[33] We have chosen to use the conceptual framework by Gilson *et al*[34] which describes the interaction between workplace and patient–provider trust to frame our analysis (see figure 1). Our purpose is to understand how trust plays out in the workplace of community health workers in South Africa, and from this make recommendations.

*Workplace trust*, defined as respectful and fair treatment in the workplace, is rooted in trust in the employing organisation, trust in your supervisor and trust in your colleagues. Patient–provider trust is rooted first in *interpersonal trust*, in this case between a patient and CHW. Patient–provider trust is also rooted in *institutional trust*—the extent to which the CHW and patient can trust that the health system will support the CHWs to act in the best interests of the patient. Here the health system refers in an immediate sense to the CHW's supervisor, the facility manager, other staff at the facility, as well as the broader health system (access to equipment, medication, effectiveness of referrals to hospital or social services). Both *interpersonal and institutional trust* are influenced by the individual provider's characteristics, their personalities, past experiences, skills, knowledge, including, for example, sensitivity to patient concerns (eg, stigma), ability to maintain confidentiality despite living in the same community and to draw support from the health system. As a result interpersonal trust always influences institutional level trust.

## METHODS
### Study design

In the initial observation phase of a 3-year intervention study in Sedibeng district, Gauteng province, we studied six CHW teams with different configurations of supervisors and locations. In analysing the observation data (reported elsewhere[35 36]), trust was a re-occurring theme. In this paper, we present the qualitative data from the four, non-intervention teams, to examine the role of trust in greater depth.

### Setting

In Sedibeng, at the time of the study, there were 39 CHW teams in 37 of the district's 72 wards (smallest geopolitical area). Sixteen of the teams were based at a health

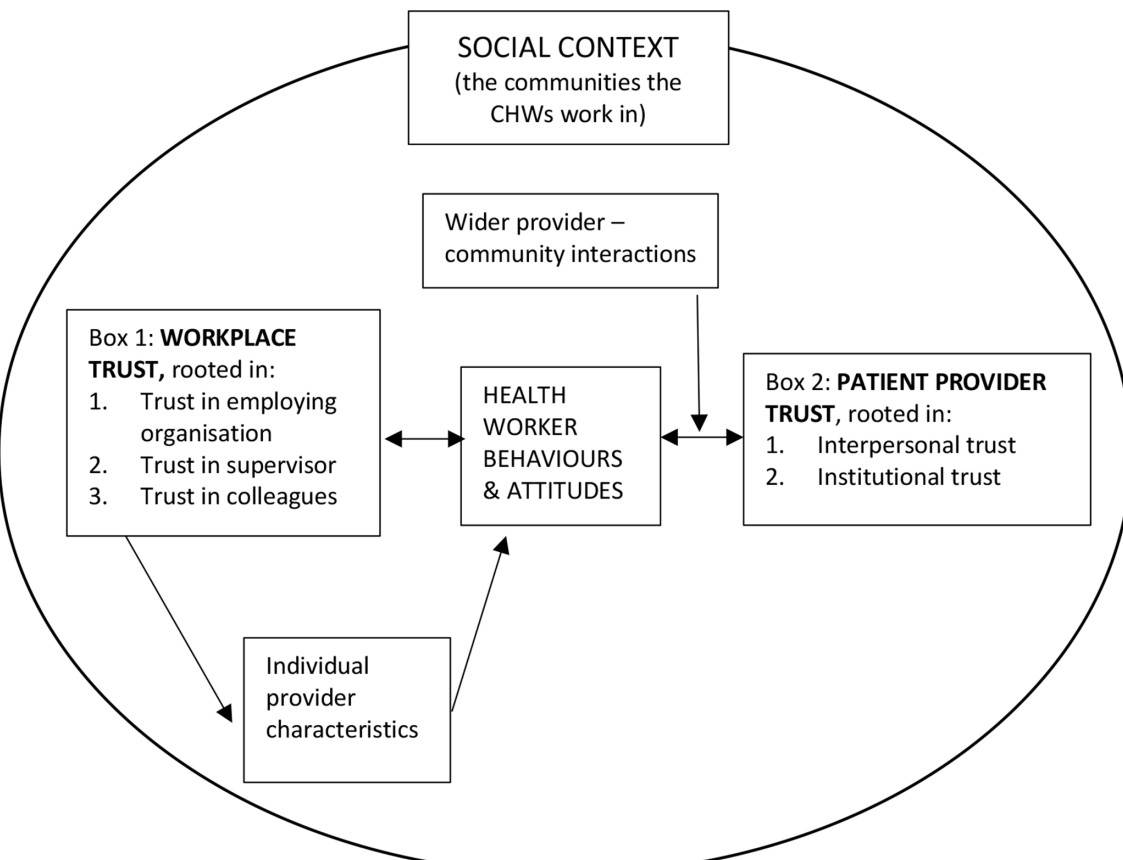

**Figure 1** Conceptual framework on trust adapted from Gilson *et al*, 2005. CHWs, community health workers.

post and the remaining 23 were clinic-based. A health post consists of one or two temporary wooden structures (providing three to six rooms), without electricity and often with irregular water supply. It is managed by one or two PNs. In addition to the scope of practice described above, in Sedibeng the CHWs deliver chronic medication to elderly, or disabled, patients.

The CHWs were not formally employed by the government. They were an outsourced workforce and paid a minimal stipend for 6 hours work per day. Two months before the start of our fieldwork, CHWs had been on strike over their conditions of employment. During the fieldwork, the CHWs were paid R2500 (ZAR), below the minimum wage (R3500 per month), however, by the end of 2018, it was increased to the minimum wage. All CHWs were from the local community.

The four teams served communities that differed by type of population and geography (see box 1). Housing was either formal brick housing, or informal shacks often made from old corrugated iron, wood and plastic. Residents in the informal settlements are often internal migrants, from poor rural provinces, or long-term migrants, often second generation, originally from Lesotho without South African identification documentation,[37] who cannot access government social grants or obtain employment.

### Data collection
Interview and observation data were collected between September 2016 and February 2017 (table 1).

Fieldworkers (FWs) were trained in research methods, ethics and the study tools using extensive role-plays and observation practice. They were also given an orientation to community-based healthcare by an experienced nurse. Interview guides and observation templates (see online supplemental material) were revised after piloting and feedback from the FWs.[36] FWs (none of whom were from the community), introduced themselves to participants as working for the study under the University of the Witwatersrand. Interviews were 15–60 min and focus group discussions (FGDs) 60–90 min in duration. No repeat interviews were conducted.

---

**Box 1    Description of the teams and the communities**

**Team 1** – A clinic-based team supervised by a professional nurse (PN) and two enrolled nurses (ENs); well-established, better-off community, with relatively low levels of unemployment. Many residents are elderly to whom the community health workers deliver medication.

**Team 2** – Team based at a health post and supervised by a PN and EN. Largely an informal settlement with very high levels of unemployment, with cases of extreme poverty and child malnutrition.

**Team 3** – Team based at health post supervised by a PN and two ENs; predominately formal brick housing, as well as an area with informal shacks.

**Team 4** – Health post team supervised by a PN and an EN; predominately government-built brick houses with back-room shacks, and a small informal settlement.

---

| Table 1 | Qualitative data collected | | | | |
| --- | --- | --- | --- | --- | --- |
| **Field site** | **Team 1** | **Team 2** | **Team 3** | **Team 4** | **Total** |
| Days of observations with CHWs (number of household visits) | 24 (88) | 18 (43) | 26 (79) | 24 (65) | 92 (275) |
| Number of focus group discussions with CHWs (44 participants) | 2 | 1 | 2 | 2 | 7 |
| Number of key informant interviews | 1 EN<br>1 PN | 1 EN<br>3 PNs<br>1 FM | 2 ENs<br>1 PN<br>1 FM | 1 EN<br>1 PN<br>1 FM | 5 ENs<br>6 PNs<br>3 FMs |
| Number of patient interviews | 14 | 6 | 12 | 15 | 47 |

CHWs, community health workers; EN, enrolled nurse; FM, facility manager; PN, professional nurse.

*Observations (obs) of CHWs*—CHWs were selected randomly by drawing names out of a hat, on the first morning of a 4-day observation period. The FWs observed CHWs at work in the community, with or without a supervisor. The FWs took detailed notes using a template.

*FGD* topics included descriptions of the types of activities carried out by CHWs and their successes and challenges.

*Key informant interviews* were conducted with the facility manager (FM), and nurse supervisors (EN or PN) to discuss typical activities, resources, how the programme ran and its successes and challenges.

*Patient interviews*—If during an observational visit a household member was given a referral by a CHW, the FW asked their permission to conduct a follow-up interview 1 month later to understand the patient's experience during any follow-up actions.

All participants were purposively sampled. All available CHWs participated in the FGDs and all nurse supervisors and facility managers were interviewed. None of the participants were known to the FWs prior to the study. All data collection was conducted in person, audio-recorded and transcribed/translated verbatim by the FWs who wrote reflective notes after each day. Patient interviews were summarised by the FW. We deemed data saturation had been reached when each pair of CHWs had been observed for at least 4 days.

### Data analysis

Raw data on trust was extracted from transcripts and notes into a data extraction table for each site by JAW (research fellow). Data was placed chronologically so that the stories flowed from day-to-day and we could connect different participant's perspectives on the same issues. Familiarity with the extracted data enabled JAW, JG and FG (all women with PhDs in public health/health science and experienced in qualitative research methods) to develop a common understanding about trust; based on this understanding, JAW identified a range of conceptual frameworks, and we chose Gilson *et al*[34] as having the closest fit with the data.

We manually conducted a thematic analysis of the extracted data by drawing out the data on trust, under related themes informed by the framework, into a second data extraction table.[34] We compared data segments and quotes under each theme across the sites and between the participants in an iterative process of reflection. This allowed us to then organise data under each form of trust, as it is presented in the results. We ordered quotes from broad to narrow issues under each theme as this enabled us to develop the logic of our argument.

Participants did not provide feedback on the findings, but one FW was engaged to help check the interpretation of data. We present the results as a narrative including anonymised participant quotations. We draw quotations from across all field sites/teams and across participant groups.

### Patient and public involvement

CHWs, patients and the public were not involved in the design, conduct, reporting or dissemination plans; however, facility, district and provincial managers were involved. We have conducted feedback sessions with CHW teams, facility, district and provincial managers for the study as a whole.

### RESULTS
### Workplace trust
#### Conditions of employment

Many CHWs complained the stipend was not adequate to meet their basic needs and wanted fairer contracts with 21 days of annual leave, and employment after 50 years of age: "We feel that the Department of Health failed us. We were promised that we would be employed permanently, that our stipend would change to a salary". (CHW_FGD2-Team4) During the strike before the fieldwork: "They started toyi-toying [protest dancing], singing the swearing songs at us [staff]. It was just havoc". (FM-Team2) At times, the strike was violent, corroding trust: "The CHWs attacked us and some of the staff were injured". (FM-Team2) Some CHWs continued a strategy of passive resistance: "They did not want to go to the community [to work]; some mentioned that they have nothing to eat at home… one said she will not come to work until she is paid". (FW_obs_CHW-Team2)

## Working conditions—the health system

The physical conditions at the health facilities were challenging. At the health post: "The patients wait outside and when it is raining, they get wet. In the consultation rooms, the nurses put buckets because the roof is leaking". (FW_obs_CHW-Team2) At the clinic, the supervisors complained: "We need space to work. It is frustrating because we work in the kitchen". (PN-Team1) The small spaces compromised infection control and made confidentiality difficult. Where there was no space inside, the CHW team met outside, limiting the possibility of discussing patients in confidence.

CHWs work in pairs for safety, even though the CHWs know their communities well: "Sometimes you go to the house with lots of males and we are scared that we might get raped. So, our lives are at risk". (CHW_FGD1-Team4) They were not given adequate personal protective equipment against TB. Other anxieties were not wearing gloves when caring for patients with bedsores, being pricked by a needle, being harmed by dogs, or people living with mental illness.

The CHWs were not supplied with a uniform; their supervisors suggested the CHWs wore black trousers and a white top to make them identifiable and look professional. The CHWs were given work bags containing equipment, however, glucose strips or batteries for the blood pressure monitor were not routinely replaced: "It is so embarrassing when you are at the household, only to find out that the blood pressure machine is not working". (CHW_FGD1-Team3) The CHWs were told that if they should lose the equipment, they would have to replace it. Their bags were routinely searched, and the content was ticked off an inventory by the clinic security guards: "as if they were thieves". (FW_obs_CHW-Team1) One PN threatened to stop their stipend if they did not return their bags each day, even though it was not within her power to do so.

## Working conditions—the community

The CHWs are confronted with many complex, and at times, tragic situations: "The child's clinic card was lost. The shack caught fire and the family lost everything". (FW_obs_CHW-Team4) Community members were often intoxicated: "The CHWs find people sitting and drinking a Black Label [beer]. How do you talk to someone who is drinking liquor and talking nonsense?" (PN1-Team2) This made it difficult for the CHWs to do their jobs: "A patient said to us 'I am HIV positive but I don't take antiretrovirals. All I need is men and I survive'. The lady looked drunk. The CHW asked if she drinks alcohol and she said yes, she is trying to reduce stress". (FW_obs_CHW-Team3) The use of drugs was also apparent: "Some households, you find an old woman crying about her misbehaving grandson using nyaope [drugs]. You have to listen and give some counselling and leave when she is feeling better". (FW_obs_CHW-Team1) Descriptions of physical abuse were not uncommon: "The granny started crying. The CHW gave her a tissue to wipe the tears. The granny said the man we passed outside is her husband, who abused her and made her leave home". (FW_obs_CHW-Team1) In another observed visit, when asked how she is doing, a patient replied saying: "she is stressed. She showed the CHW her bruises on her arms and said her husband is beating her". (FW_obs_CHW-Team4).

Many community members did not have legal documentation. With no South African identification, children's births are not registered nor are they eligible for government social grants: "The patient does not receive the pension grant because she is from Lesotho. Her son is also unemployed. The CHWs have tried to involve social workers and the police but there is no solution because of the documents". (FW_obs_CHW-Team2) This was a common occurrence, and CHWs are often not able to assist: "Our intervention is not enough because people are not working, those from Lesotho don't have identity documents. You are breaking their hearts because you can't give them anything, besides filling in the registration form and asking: 'Is there someone with TB? Is there someone working?,' when there is no food. You can't even say I will get food parcels, there is nothing". (CHW_FGD-1-Team2) Witnessing such poverty affected the CHWs: "We carry these stories because we are also human. I wish we could have one whole day just to talk about what we have seen and observed". (CHW_FGD-1-Team1)

## Relationships with colleagues

A hierarchy played out in the health facilities, with the CHWs on the bottom rung of the ladder: "Everyone tells us what to do. We have to do everything we are told to do without any question". (CHW_FGD-1-Team1) CHWs felt their work was not valued: "We have been working here for a long time and no one recognises us. We are just a group of fools". (CHW_FGD-1-Team2) One FM expressed this hierarchy: "The nurses are trained, they know exactly what they are supposed to do, unlike CHWs; they are just called from the streets". (FM-Team2) She justified her distrust: "When we send them out, there are those who will not be doing their work, they will go to their own places to do laundry and clean their houses". (FM-Team2) Her trust had been further eroded during the strike: "It is difficult for us to interact with them because they are very same people who had attacked us. We talk for the purposes of work, other than that, we are keeping them at arm's length". (FM-Team2) Another FM said that despite her grievances about them, "we learn to live with them and realise that these people are useful to us". (FM-Team4)

However, the CHW's supervisors generally praised them. One EN talked of leading the CHWs, even while calling them children: "I know my children [CHWs]. It is important, you must be friendly and polite. Don't be a boss, be a leader. I talk to the CHWs. I tell them 'Be friendly… as I am'". (EN1-Team1) This attitude created a two-way relationship: "The CHWs are comfortable to raise issues with her. If they answer her, 'No sister [nurse], this should be like this and that', she listens to

them, and allows them to take initiative and solve the problems". (FW_obs_PN- 1-Team4) Some of the ENs and PNs worked hard to reduce social hierarchy at the clinic, and ensure the CHW abilities were used constructively: "I feel that CHWs are not appreciated. I took my time and studied each CHW. Now I know how to handle each of them. If a CHW is rude, I bring her close to me and tell her that you will be doing statistics with me [collating activity data]". (PN1-Team1) This PN created leadership opportunities: "I said to her [CHW] you are their leader, if they have a problem, they must address it with you [CHW] and you will tell me [PN]". (PN1-Team1) Another PN tried to help the CHWs understand the importance of confidentiality and trusting relationships with patients: "Sometimes, they gossip about their patients and that condemns the family. If they identify a TB patient and I would ask them to adopt that family". This was not always successful "[however] when I ask if they have delivered the treatment to the patient, [the CHW responds] 'why doesn't the patient come and fetch it?'"." (PN2-Team2)

Some supervisors viewed the CHWs as "unreliable. They just work for the sake of money… they aren't doing it wholeheartedly". (PN1-Team2) This PN did not trust that they recorded the patient's blood pressure results accurately saying: "[CHWs] are not so honest." (PN1-Team2) One PN would belittle the CHWs: "She [PN] even shouts at us in front of the patients. We are adults, mothers and we have our own houses… she does not respect us". (CHW_FGD-1-Team3)

The less experienced ENs struggled to gain the respect of the CHWs: "I think it is because of my age maybe. They see me as young even though I am a nurse". (EN1-Team2) The CHWs relied on the PN, dismissing the EN's help: "The PN is much better, she was able to sit down with us and show us where we went wrong. The EN does not even sit with us. Last time there was a fight here they told her that, 'you are distant from us'". (CHW_FGD-1-Team2) The PN was aware this was a problem: "She [EN] has no comradeship with them. I can see that there are some [CHWs] who want to disrespect her because she is still very young". (PN1-Team2). One EN had little experience, and the CHW had to teach her: "The ENs know nothing about WBOT. We are the ones who had to teach them but they are getting more salary than us. That is not fair; that is why we get angry and strike". (CHW_FGD-Team4)

However, there was evidence of teamwork between the EN and CHWs: "We have two ENs who are able to walk to the field with us. They are helpful in intervening in our cases". (CHW_FGD-1-Team1) Another EN was happy to follow the CHWs' routine, helping when needed: "I don't change whatever they're doing that day, because they know their patients. I go and supervise wherever they are going. If they are having any problems, they will take me to those places so that I can help". (EN1-Team3) The supervisors were key to enlisting the support of other social services, even if their help was limited (Vignette 1).

### Vignette 1

"I told the police there was a house written, 'no entry'. The police came and I got into the van and went with them. We found an old lady. There was a small bundle; she was sleeping on some blankets. It was winter. There was bread, a towel, and porridge with nothing else. We bathed her and put on body lotion. After she bathed, she sat in the sun… she said, 'you remind of the days I used to bathe like this'. I gave her a blanket, towel, underwear, and nappies. I sat down and phoned SASSA [The South African Social Security Agency], and Home Affairs. When they checked her records, she was deported in 1975 back to Lesotho but she found a way to return to South Africa. Home Affairs said there is no way this woman can get an ID book. I am telling you about the problems the WBOT are experiencing." (PN2-Team2)

### Interpersonal trust (patient–CHW)

There was a huge appreciation and respect for the work of the CHWs. "These people found me dying… I was not drinking water, not eating. The CHWs came every morning; they are ever-caring". (FW_obs_PN-Team4) Relying on the CHWs for emotional support was common: "I have found people that I can pour out my problem to… I feel very good after talking to them". (Patient-Team3) One man noticed that: "…we [CHWs] sit outside in the sun and he tried to erect a small shack so that we can sit there". (CHW_FGD-1-Team2) When the CHWs were on strike: "The community elders were even saying: 'if we were able to walk, we were going to join you [on the strike]'". (CHW_FGD-1-Team1) Gaining respect, and being acknowledged as a nurse, was hugely motivating: "I feel tall especially when they call me at the [shopping] mall saying 'Sister!'. Wow, I feel good'". (CHW_FGD-1-Team1)

Generally, the CHWs were sensitive to people's personal matters: "We do not tell the mother that her daughter did the HIV test. We can only be free to talk about it if the mother initiates the topic". (CHW_FGD-2-Team1) Home visits provided the opportunity to build supportive relationships: "I started working with the patient during her pregnancy after she was diagnosed with HIV/AIDS. She was devasted and unfortunately miscarried because of stress and depression. I continued to visit, and we developed a good relationship. The patient is taking treatment very well and her CD4 counts have improved". (FW_obs_CHW-Team4) On occasions, CHWs and nurses were not sensitive enough. "The EN asked the teenager if she is sexually active. The girl found the question very difficult to answer in front of her grandmother. One of the CHW's advised the EN to speak with the teenager in a private space. The EN took the advice and used the nearby kitchen to talk with the teenager." (FW_obs_CHW-Team1) The teenager was invited to the health post where she received sex education and contraceptives.

Some community members did not trust the CHWs. The PN supervisor had to reassure a patient that an HIV test would not be done by a CHW. Sometimes people do not answer their doors: "… because they fear that people would think they are HIV positive, but we are visiting

everyone". (PN1-Team4) They were blamed for not delivering medications on time: "We go to their houses to deliver medications and they are not there. Then the patient will come to the clinic to complain. So, it seems as if we don't do our work". (CHW_FGD-1-Team1) Occasionally, when the clinic does not have any medication, the CHWs were blamed for selling on the medication.

## Institutional trust (patient–CHW–nurse–health system)

CHWs can make a formal referral to the health facility to obtain care for their patient. In two sites, referred patients did not have to queue at the clinic, but could go straight to the CHW supervisor—this increased the patient's institutional trust of the health system. The teenager's referral (above) enabled a streamlined visit for her first contraception visit: "The referral helped me because it was the first time I visited the clinic. It made it easy as I knew exactly who I was looking for". (Patient-Team1) However, without adequate support from the clinic, CHWs often struggled to support patients: "I had an incident where a patient who tested negative [HIV] throughout her pregnancy but on delivery tested positive. The clinic gave her antiretrovirals but she left them at the clinic. I took them to the woman and she told me, she won't take those them because she is not HIV positive. I had to do counselling which I am not qualified to perform". (CHW_FGD-Team4) The CHW efforts are sometimes frustrated by the clinic 'rules': "Some households do not have food to eat and the patient is on treatment. I contribute something [from my own pocket] so that the patient can eat, but when I ask for porridge from the clinic, the nurse tells me that I have to bring the patient [to the clinic]. That is a big problem because the patient is sick and has no transport money. So, yah it hurts". (CHW_FGD-1-Team3) The nurse was not willing to accept the CHW's word that the person cannot come, and failure to support a patient reduces the CHWs credibility in the community.

In other instances, there was a breakdown in communication with other parts of the system: "She used to go and collect her [government] grant by herself but for the previous 2 months she has been bedridden. We promised her a wheelchair. We phoned the hospital, but it was not possible for us to take the old lady to the hospital because of the distance. I don't know what happened [after that]". (FW_obs_CHW-Team2) At times, the complex transitory nature of people's lives made helping them difficult: "A woman was breast feeding and defaulted on her antiretrovirals, and was refusing to come to the clinic, so they were concerned about the safety of the baby. The CHW referred the patient to the social worker, but she does not think the social worker managed to find the patient because she disappeared". (PN3-Team2)

Some nurses were sensitive to the challenges that the communities faced. "When you sit down and talk with them you would find that they have buckets of stressors. I don't have tablets for stress. I have to try and talk to the person". (PN1-Team2) The same nurse was also aware that her privileged position made her insensitive to patients' challenges: "I asked this woman whether she had bathed or not, because the child was dirty and the woman was also untidy, and she said no. I really got embarrassed because she told me that there was no body soap because there was 'no one working at home, we are left with our grandmother, our mother is dead, and we are from Lesotho'"." (PN1-Team2)

However, negative attitudes of nurses often affected the patient's willingness to attend the health facility: "The CHWs asked an old lady why she is not taking her treatment. The lady said the sister [nurse] at the clinic doesn't talk to her well, so it is better for her to stop going to the clinic". (FW_obs_CHW-Team3) Some patients confided in the CHWs about the nurses' behaviour: "[the patient] said if the nurse comes to their houses and speaks the way she speaks when she is at the clinic, they are going to hit her". (CHW_FGD-2-Team3) Vignette 2, describes a woman who was no longer taking her antiretroviral treatment due to a disagreement with the health post staff. In vignette 3, a nurse is rude to a vulnerable pregnant woman who withdraws from care; the CHW enlists the PN's support to get the nurse to apologise but fails.

Responsible for finding defaulting patients, the CHWs have to find ways to repair the patient's trust in the health system: "We talk to the patients, encourage them to go for the sake of their health and ignore the nurses'

> **Vignette 2**
>
> The CHW asked the woman why she is breast feeding while she is HIV positive and not taking treatment: "She said she and her child are both fine, the child has never been sick, and she is picking up weight: 'I will see about that thing when I am sick'". (FW_obs_CHW-Team4) The nurse had taken bloods from her baby, but the results were lost. Before retesting the child, the woman was insisting: "The clinic must first tell me what happened to the blood that was drawn". (FW_obs_CHW-Team4)

> **Vignette 3**
>
> A patient was raped as a teenager and contracted HIV. Her first child passed away and her second child was also HIV positive. The patient usually takes antiretrovirals. She is pregnant. When visiting the clinic, a nurse refused to check the unborn baby because the mother did not have a transfer letter from her previous clinic in the Eastern Cape. The consultation ended with the nurse saying: "I don't care if you give birth in the toilet or in the street. It is not my problem". (FW_obs_CHW-Team3) Feeling angry and upset, the pregnant woman refused to go back to the clinic. With tears flowing down her face the woman said: "I know the rules of the clinic for a pregnant mother who is HIV positive, and I follow them always. I did not choose to be HIV positive". (Patient-Team3) The CHW promised to collect her medication on her behalf, and also she would report the case to the supervisor, saying: "she [patient] must not worry as she is not the first one she has treated badly and this time she [CHW] is going to report her". (FW_obs_CHW-Team3) However after the patient and nurse communicated, the patient reported: "the sister [nurse] denied all the things she said to me, they wrote on my clinic card that I will never come back to this clinic, and the clinic will not be accountable should anything happen to me"." (Patient-Team3)

attitude". (CHW_FGD-2-Team3) However, the CHWs could not guarantee patients will be treated better next time: "I [CHW] told the patient that when you [don't go to] the clinic, you are killing yourself as you are the one who is taking treatment, not the sister [nurse] and the life that you are living is yours but not the sister's". (CHW_FGD-1-Team3) A few people said they would only return to seek care if the CHW accompanied them: "Okay, I will come only if you [CHW] will be there too". (FW_obs_CHW-Team4)

## DISCUSSION

In this paper, we have explored both workplace trust, patient–CHW interpersonal trust and patient–health system (institutional) trust, among CHW teams in South Africa. The CHWs are the lowest cadre in the health system; their conditions of employment, their working environment, the lack of necessary equipment to perform their work safely and the treatment by some colleagues indicated to the CHWs that their work was unrecognised, their contribution untrusted. The low levels of monetary incentives and poor working conditions of CHWs have been reported elsewhere in South Africa,[38] and other resourced constrained settings.[39] By striking, the CHWs in our study were demanding better pay and employment conditions, as well as recognition of the importance of their work.

The CHWs are working in communities mired in complex social problems the result of long-term structural poverty. Dysfunctional family relationships impact on patients' ability to look after themselves and take their treatment. Alcohol abuse blights the lives of community members, making the work of the CHWs harder. The lack of identification documents among long-term migrants leads to desperate poverty that makes accessing healthcare difficult, even with the CHWs' efforts. Some supervisors understood the extent of the poverty in the surrounding communities, from which the CHWs themselves came. These supervisors worked hard to overcome the social hierarchy by building the CHW's skills, and supporting their efforts to help the community.

There was considerable evidence of interpersonal trust between patients and CHWs, with many people appreciative of their work. CHWs operate in a unique environment, where household visits enable strong relationships to be built, but living in the same community can test the CHWs' ability to maintain confidentiality.[10] CHWs have to make sensitive judgements about when and what to ask people in order to build trust, a difficult terrain to navigate, particularly because of the vulnerability of many of their patients.

The attitude of some of the facility-based nurses ('bad apples'[40]) led some patients to withdraw from care.[41] Insensitive to stigma and barriers to accessing care that the socio-economic conditions of people's lives create, nurses' behaviour offended patients. CHW's attempts to repair trust often failed due to the vulnerabilities of the community, and lack of support from the health system, underpinned by poor workplace trust including CHWs often fraught relationships with their colleagues. We have reported elsewhere that inadequate and unpredictable support from the clinic negatively affects the CHW's ability to provide care and in turn, their credibility in the community.[35]

Migrants, often unable to seek care because of their poverty, are being denied a fundamental human right. Section 27 of the 1996 South African Constitution states that 'everyone has the right to have access to health services'. The National Health Insurance bill[42] states that illegal and undocumented migrants 'will receive basic healthcare services' (emergency care and treatment for HIV, TB and malaria), but not general primary healthcare or sexual and reproductive services.[43] It is not possible to provide effective HIV treatment without related primary healthcare services,[44] and this is at odds with the SDG of 'leaving no one behind'[45] and achieving universal health coverage. Societies need to care for everyone in them.

### Strengths and limitations

A strength of this research is the number of observational days that the FWs spent with the CHWs and other staff across the four sites to understand their daily work and community interactions. The resulting rich data illuminates the reality of the vulnerable communities lives in which CHWs work in—desperate poverty, alcoholism and gender-based violence, as well as the relationships, and so trust, between the community, community healthcare workers, their colleagues, supervisors and in the health system. As the data was not collected explicitly for research on trust, participants may not have expressed all their views on trust. Moreover, text describing relationships, based on individual perceptions, is often difficult to interpret—we suggest that our detailed knowledge of the context, and our rigorous analysis process described above, has ensured that our interpretations are true to our participants' experiences.

### Recommendations

Given the interconnected nature of workplace, interpersonal and institutional trust, our recommendations include:

1. CHWs and nurses should be provided with opportunities to develop a better understanding of, and empathy for, the community's health and social situation.
2. Facility managers and nurses need to work to overcome social hierarchy in the facility, so CHWs feel supported in their workplace and patients feel cared for.
3. Inexperienced ENs need to be mentored while they develop as CHW supervisors.
4. In communities with complex social problems, the CHWs and their supervisors need strong intersectoral collaborations with other services.
5. Migrants need to have the right, and means, to be able to access care.

## CONCLUSION

CHWs' role in enabling vulnerable communities to access care is underpinned by workplace, interpersonal trust and institutional trust. Without these different forms of trust, CHWs struggle to assist patients to stay in care; yet creating trust in marginalised communities struggling with structural poverty is far from easy. Nurses and CHW supervisors need to be sensitive to the hierarchy created by social inequalities and the barriers that patients face in accessing care. They need to support the CHWs in helping patients overcome these barriers. The government's role in ensuring migrants' rights to accessing healthcare services is crucial in developing trust.

**Acknowledgements** The authors would like to thank the Sedibeng community and primary healthcare staff for being involved in the study. Thank you, also to the dedicated team of fieldworkers working for the Bathlokomedi project.

**Contributors** JG and FG are the principal investigators and award holders on this grant, conceptualising the study and managing data collection. They contributed to the supervision, data analysis and interpretation and drafting this article. JAW extracted and analysed the data and wrote the article, with JG editing drafts. All authors reviewed the final draft of this article.

**Funding** The work was supported by the UK Medical Research Council (grant number: MR/N015908/1).

**Competing interests** None declared.

**Patient consent for publication** Not required.

**Ethics approval** All participants gave informed consent. The project received ethical approval from the University of the Witwatersrand Human Research Ethics Committee (Medical) (M160354), the Gauteng Provincial Health Research Committee and from the University of Warwick Biomedical and Scientific Research Ethics Sub-Committee (REGO 2016–1825).

**Provenance and peer review** Not commissioned; externally peer reviewed.

**Data availability statement** Data are available upon reasonable request.

**ORCID iDs**
Jocelyn Anstey Watkins http://orcid.org/0000-0003-4984-1057
Frances Griffiths http://orcid.org/0000-0002-4173-1438
Jane Goudge http://orcid.org/0000-0001-6555-7510

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
