## [Reviewer comments · BMJ Open]

ARTICLE DETAILS

TITLE (PROVISIONAL)	Community health workers' efforts to build health system trust in marginalised communities: A qualitative study from South Africa
AUTHORS	Anstey Watkins, Jocelyn; Griffiths, Frances; Goudge, Jane

VERSION 1 – REVIEW

REVIEWER	Sozinho Ndima University of Eduardo Mondlane, Mozambique
REVIEW RETURNED	02-Oct-2020

GENERAL COMMENTS	In general the manuscript is well structured and presented. I would suggest to insert a new column for the total number of interviews conducted in the table 1, in page 5.
--

REVIEWER	Peijia Zha Rutgers, the State University of New Jersey USA
REVIEW RETURNED	14-Nov-2020

GENERAL COMMENTS	Overall, this is a clear, and concise manuscript. But it needs much more work to be a good contribution to the how the CHWs building the trust relationships in the underserved community, why the trust is so hard to build in this specific environment. There are many weaknesses to this paper. Especially in discussion section, its reporting of workplace trust, interpersonal trust between the patient and CHW, and the institutional trust patients place in the health system without taking into account the ways in which various trust are inter-related, for example, trust is an ambiguous concept, the interpersonal trust at individual level always intervening institutional level trust. In addition, there were few grammatical errors throughout the manuscript, please be meticulous to review and proofread the paper from top to bottom. Specific Comments: Background: 1. more information are needed for CHW, who they are, where they got recruited, what are their qualifications for CHW position, and what kind of trainings they got from department of health or nurses. Data analysis: a thematic analysis approach was used to analysis the data, more clear and detailed steps doing this analysis need to be addressed. Discussion: I would have liked to see a much richer discussion section that explains your key theoretical contributions. It would be helpful to discuss in greater detail the specific outcomes why the trust is not building at both individual level and institutional level, and how the health workers behaviors and attitudes impact the trust relationship at both levels.
--

	Limitation: study locations is a limitation for generality of this study. However, I would like suggest the author talk about research methodology, trust is a vital concept and much based on individual perceptions, which is very difficult to interpreted. Therefore, neither qualitative studies nor quantitative studies alone can provide generalizable, objective evidence.
--	--

VERSION 1 – AUTHOR RESPONSE

Reviewer: 1

Dr. Sozinho Ndima, Eduardo Mondlane University

Comments to the Author:

In general the manuscript is well structured and presented.

I would suggest to insert a new column for the total number of interviews conducted in the table 1, in page 5.

Response: Thank you Reviewer 1. We have now inserted a column into table 1, with the total number of interviews, PAGE 5.

Reviewer: 2

Dr. Peijia Zha, Rutgers University

Comments to the Author:

Overall, this is a clear, and concise manuscript. But it needs much more work to be a good contribution to the how the CHWs building the trust relationships in the underserved community, why the trust is so hard to build in this specific environment. There are many weaknesses to this paper.

Especially in discussion section, its reporting of workplace trust, interpersonal trust between the patient and CHW, and the institutional trust patients place in the health system without taking into account the ways in which various trust are inter-related, for example, trust is an ambiguous concept, the interpersonal trust at individual level always intervening institutional level trust.

Response: Thank you Reviewer 2. We agree, trust is a rather ambiguous concept, although we would argue that existing definitions and theoretical frameworks have done much to elucidate it. We have added this as a sentence on PAGE 4, including a citing a new reference. We have carefully described the framework that we have chosen to use in the background section. In this section, we have added a sentence to explain that interpersonal trust at individual level is always interacting with institutional level trust on PAGE 4.

In addition, there were few grammatical errors throughout the manuscript, please be meticulous to review and proofread the paper from top to bottom.

Response: We have read and re-read the paper very carefully to identify the grammatical errors. To our knowledge we have identified all of them.

Specific Comments:

Background:

1. more information are needed for CHW, who they are, where they got recruited, what are their qualifications for CHW position, and what kind of trainings they got from department of health or nurses.

Response: We have now added the information on who the CHWs are, where they got recruited, what

are their qualifications for CHW position, and what kind of trainings they got from department of health or nurses, see PAGE 3.

Data analysis: a thematic analysis approach was used to analysis the data, more clear and detailed steps doing this analysis need to be addressed.

Response: We have described the steps involved for the thematic analysis in more detail, see PAGE 6.

Discussion: I would have liked to see a much richer discussion section that explains your key theoretical contributions.

Response: We are actually of the view that the Gilson et al. framework is adequate for our purpose. The key concepts - workplace trust, interpersonal trust and institutional trust are what we are concerned with. Therefore, the contribution of our paper is the richness of the data (now explicitly written on PAGE 12), the depth of understanding of the nuances of the relationships in a particular setting, and the difficulties in generating trust, particularly in poor communities, rather than addition to the theory per se (in the discussion). We hope you agree.

It would be helpful to discuss in greater detail the specific outcomes why the trust is not building at both individual level and institutional level, and how the health workers behaviors and attitudes impact the trust relationship at both levels.

Response: Our results do show examples of trust at individual level (between CHW & supervisor, and community/patient & CHW). However, we also provide much of evidence of where trust does not exist. The key reasons are (and how we have structured our results): a) the CHW's working conditions, including social hierarchy within the health system, and the desperate poverty in the community; and, b) the lack of support from both the CHW's colleagues and from health system structures in ensuring that the patients receive the care they needed. While in the results we provide evidence of these conclusions, in the discussion we distil these two key points, and in the list of recommendations we set out the key issues that facility/ district/ provincial managers need to focus on, in order to generate trust. Given the constraints of the word limit, we hope this is sufficient.

Limitation: study locations is a limitation for generality of this study. However, I would like suggest the author talk about research methodology, trust is a vital concept and much based on individual perceptions, which is very difficult to interpreted. Therefore, neither qualitative studies nor quantitative studies alone can provide generalizable, objective evidence.

Response: We agree that trust is a vital concept, based on individual perceptions, and is difficult to interpret. We have edited this section on PAGE 12 (and the article summary on PAGE 2). In the limitations section, we have also added the following sentence:

“Moreover, data describing relationships, based on individual perceptions, is often difficult to interpret – we hope that our detailed knowledge of the context, and our rigorous analysis process described above, has ensured that our interpretations are true to our participants’ experiences.”

Reviewer: 1

Competing interests of Reviewer: None declared

Reviewer: 2

Competing interests of Reviewer: None Declared

VERSION 2 – REVIEW

REVIEWER	Peijia Zha Rutgers, the State University of New Jersey USA
REVIEW RETURNED	22-Feb-2021
GENERAL COMMENTS	All required revision were addressed. It can be accepted and published.